environmental science/bioengineering/health and disease and epidemiology

noise pollution, parasitic disease, host–parasite dynamics, animal welfare

**Author for correspondence:**
Jo Cable
e-mail: cablej@cardiff.ac.uk

# Noise pollution: acute noise exposure increases susceptibility to disease and chronic exposure reduces host survival

Numair Masud[1], Laura Hayes[1], Davide Crivelli[2], Stephen Grigg[2] and Jo Cable[1]

[1]Schools of Biosciences, and [2]Engineering, Cardiff University, CF10 3AX Cardiff, UK

NM, 0000-0002-9561-6494; JC, 0000-0002-8510-7055

Anthropogenic noise is a pervasive global pollutant that has been detected in every major habitat on the planet. Detrimental impacts of noise pollution on physiology, immunology and behaviour have been shown in terrestrial vertebrates and invertebrates. Equivalent research on aquatic organisms has until recently been stunted by the misnomer of a silent underwater world. In fish, however, noise pollution can lead to stress, hearing loss, behavioural changes and impacted immunity. But, the functional effects of this impacted immunity on disease resistance due to noise exposure have remained neglected. Parasites that cause transmissible disease are key drivers of ecosystem biodiversity and a significant factor limiting the sustainable expansion of the animal trade. Therefore, understanding how a pervasive stressor is impacting host–parasite interactions will have far-reaching implications for global animal health. Here, we investigated the impact of acute and chronic noise on vertebrate susceptibility to parasitic infections, using a model host–parasite system (guppy–*Gyrodactylus turnbulli*). Hosts experiencing acute noise suffered significantly increased parasite burden compared with those in no noise treatments. By contrast, fish experiencing chronic noise had the lowest parasite burden. However, these hosts died significantly earlier compared with those exposed to acute and no noise treatments. By revealing the detrimental impacts of acute and chronic noise on host–parasite interactions, we add to the growing body of evidence demonstrating a link between noise pollution and reduced animal health.

# 1. Introduction

With species loss occurring 1000 times above the background rate of extinction, there is an urgent need to understand how anthropogenic activity influences ecosystem biodiversity and animal welfare [1]. Anthropogenic noise is a global pollutant. It has marked impacts on human health, from reduced cardiovascular function [2–5] to elevated cortisol levels and disrupted sleep patterns [6,7]. Indeed, from long-term cross-sectional surveys, people report a significant reduction in their quality of life when subject to chronic noise [7]. Stress responses to sound pollution have also been shown in non-human vertebrates (reviewed in [6]). Bird communities, such as the greater sage-grouse (*Centrocercus urophasianus*), have elevated faecal corticosteroid metabolites and show a decline in male lek attendance when exposed to chronic and intermittent noise [8,9]. Reproductive behaviour, including anuran mate calling, is affected by chronic roadside noise, with frogs for example having to increase song pitch leading to greater energy expenditure [10]. More than any other vertebrate system, mouse models have demonstrated that noise can impact behaviour, reproduction, metabolism, the cardiovascular system and immunology (reviewed in [11]). Even invertebrates are not exempt from the detrimental impacts of noise pollution [12].

For aquatic organisms, including fish, the potential impact of noise pollution has only recently gained attention, and this is linked to the significant rise in underwater sonar, pile driving, seismic activities and motorized vehicle activity [13]. Freshwater fish in particular are a global welfare concern, recognized as the most endangered group of animals on the planet [14,15], in addition to being a major source of animal protein for human consumption [16]. Multiple fish species have displayed primary (e.g. cortisol production; [17]), secondary (e.g. cellular immune response; [18]) and tertiary level impacts (e.g. potential disease resistance; [19]) of noise exposure. However, while there have been investigations on a range of tertiary level impacts of noise exposure on fish (e.g. [20,21]), limited work exists on disease resistance in particular.

Typically, animal species respond in one of three ways to noise exposure: (i) no apparent response to the sound stimulus (e.g. [20]), (ii) an initial stress response followed by acclimation (e.g. [21]), or (iii) consistent long-term detrimental health effects (e.g. [12]). Reduced resistance to transmissible disease is arguably the most significant long-term welfare concern of noise exposure. This is because if left untreated and/or immune suppression occurs, transmissible disease will impact primary and secondary stress responses and ultimately cause mortality. To date, only two animal studies have assessed the impact of noise on susceptibility to infections [19,20]. Of these, only Wysocki *et al*. [20] demonstrated that rainbow trout (*Oncorhynchus mykiss*) appeared unaffected by chronic eight-month noise exposure and subsequent *Yersinia ruckeri* inoculation. Parasites causing transmissible disease are recognized as one of the most significant causes of economic loss, due to host mortality in global animal trade (see [22] and [16]). For industries such as aquaculture, infectious disease has reached crisis status exacerbated by neglected stressors that compromise host immunity [23]. Therefore, the functional importance of stressors such as noise and its relation to disease resistance extends to impacts on valuable human resources.

The guppy–*Gyrodactylus turnbulli* host–parasite system has been used to understand how anthropogenic stressors impact disease resistance (e.g. nitrate enrichment: [24], animal transport: [25]). This model allows us to monitor individual infection trajectories in real time, which we do here to assess how acute and chronic noise exposure impacts resistance to transmissible disease. The host is a globally important freshwater fish, the Trinidadian guppy, that is an established eco-evolutionary model (e.g. [26]). The genus *Gyrodactylus* is a group of hyperprevalent monogenean ectoparasite species of ecological and aquaculture importance [27–29]. These so-called 'Russian-doll killers' employ progenesis and hyperviviparity allowing parasite numbers to exponentially rise threatening host survival (reviewed in [26]). *Gyrodactylus turnbulli* is a primary parasite of guppies and of major concern in the ornamental trade [30]. As the conservation status of freshwater fish is critical [31], understanding how anthropogenic noise impacts their resistance to transmissible disease is extremely timely.

# 2. Methods

## 2.1. Host and parasite origins and maintenance

Mixed strain ornamental guppies (*Poecilia reticulata*, $n = 2000$) were purchased and transported from GuppyFarm UK to Cardiff University in September 2018. All fish were ectoparasite free on arrival,

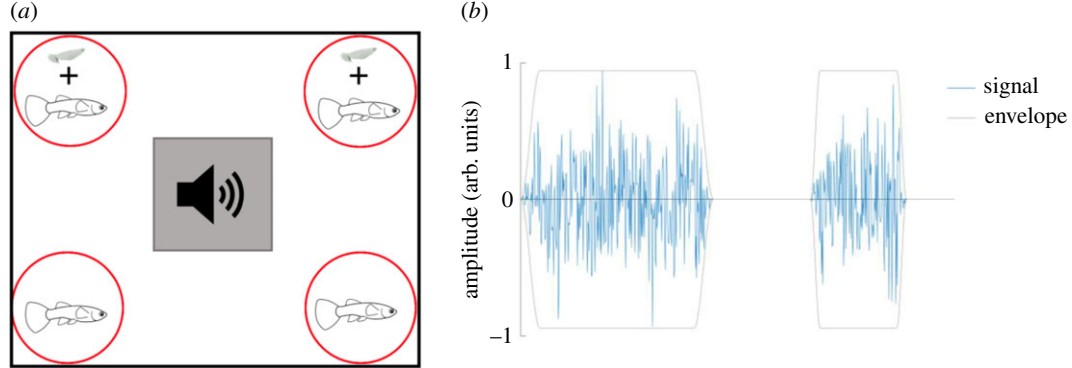

**Figure 1.** (a) Schematic of general experimental design. Guppies were exposed to one of three treatments: acute noise ($n = 24$), chronic noise ($n = 28$) or no noise, controls ($n = 52$). This was followed by experimental infections of half the fish with *Gyrodactylus turnbulli* parasites (shown here as grey worms, not to scale). Sound treatment design shown here, the black rectangle represents a glass tank ($60 \times 30 \times 30$ cm³) with an underwater speaker (grey filled square; turned off in the no noise controls). Each red circle with a female guppy represents 1 l containers in which hosts were isolated for the duration of acute and chronic noise exposure as well as control treatment. (b) White noise enveloped (i.e. turning a continuous sound into bursts of shorter sounds of random length, followed by silence of random length) to generate 'bursts' of noise that was used for both the acute and chronic treatments.

confirmed through three consecutive screens using a dissecting microscope with fibre optic illumination [32]. For experimental infections, the Gt3 strain of *Gyrodactylus turnbulli* was used, which originated from a single worm isolated from an ornamental guppy in 1997. This parasite population has since been maintained in culture pots containing at least four naive fish collectively infected with a minimum of 30 worms. Naive guppies are added to the culture when worm numbers decrease, and heavily infected fish removed (treated) and replaced to prevent parasite extinction [29]. All fish were maintained at $24 \pm 1$°C under a 12 h light/12 h dark lighting regime and fed a daily diet of tropical flakes (Aquarian®) along with freshly hatched *Artemia* nauplii every alternate day. Experimental fish were size-matched adult females (standard length (SL) range 14–27 mm).

## 2.2. Experimental design: acute and chronic noise exposure

To investigate how noise exposure impacts fish resistance to parasitic infections, guppies were allocated to either acute noise (24 h, $n = 24$, SL range 16–25 mm) or chronic noise (7 days, $n = 28$, SL range 14–21 mm) treatments prior to parasite exposure. For each treatment, control fish (acute, $n = 24$; chronic, $n = 28$, SL range 14–27 mm) were placed in identical conditions but with no noise exposure. The experimental set-up for both acute and chronic treatments (figure 1a) involved placing individual guppies in 1 l containers within a glass tank ($60 \times 30 \times 30$ cm³) equidistant from an omnidirectional underwater speaker (UW-30, Illuminate Design Ltd, Witham). Each tank consisted of four 1 l containers per speaker (acute noise = six replicates; chronic noise = seven replicates). The water level in the tanks was just below the rim of the 1 l containers and sufficient to ensure full submergence of the speakers. This host isolation was necessary to monitor individual infection trajectories, as *G. turnbulli* can directly transfer between conspecific fish upon contact [28]. To maintain water quality, all experimental fish in 1 l containers underwent complete water changes every alternate day. Underwater speakers were connected to an amplifier and subsequently a laptop to deliver the same sound file into each experimental tank. The speaker played random, intermittent white noise covering the 100–10 000 Hz range [33]. These noise files were generated using VCV Rack, an open-source additive synthesis software (https://vcvrack.com/), and then randomly enveloped (figure 1b) to generate individual 'bursts' of sound between 0.1 and 10 s, interleaved with the silence of the same random duration range. In the control ($n = 13$) tanks, the speakers were turned off and disconnected from the main power source. We note, however, the possibility of a confound in relation to the control fish not being exposed to magnetic fields. This is due to electrical currents creating a variable magnetic field to which the voice coil in the underwater speaker responds to generate sound. No noise was transmitted between the noise exposure and control tanks and the same noise levels were recorded within each 1 l container in each experimental tank, confirmed through hydrophone (Reson TC 4013) recordings and data acquisition system (Picoscope 5443B). The white noise emitted from the speaker was altered by reflections due to tank geometry, and the mechanical characteristics of the

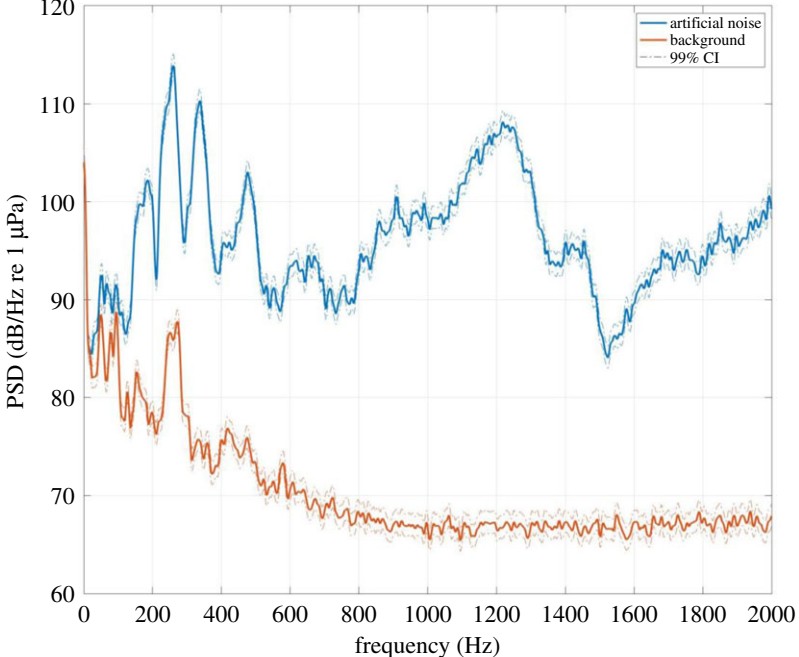

**Figure 2.** Power spectral density of the noise hosts were exposed to compared with the background noise inside a tank. Average of 10 distributions over 1 s intervals (frequency resolution = 1 Hz) and 99% CI.

medium, the tank wall and tank contents. Figure 2 shows the resulting power spectrum, measured at mid-depth of the 1 l fish containers and averaged over approximately 10 s. While we are aware that fish can respond to particle motion [34], we could not measure this as we were unable to source a suitable accelerometer. However, these sound pressures are in line with mild sound levels recorded in concrete raceways, earthen ponds and indoor aquaculture systems [35].

## 2.3. Experimental infections

Guppies were experimentally infected after acute (24 h) or during chronic noise exposure (day 7) (figure 3). For the chronic noise treatment, hosts that were infected with parasites continued to experience noise during infection trajectories. Thus, chronic exposure fish experienced noise for a total of 24 days. Experimental infections involved lightly anaesthetizing individual guppies with 0.02% MS-222, and each fish was infected with two gyrodactylid worms. Parasite transfer was conducted following standard methods of King & Cable [36]. Briefly, two worms from heavily infected donor fish were transferred to the caudal fin of recipient hosts by placing the anaesthetized donor fish in close proximity to an anaesthetized naive host, monitored continuously using a dissecting microscope with fibre optic illumination. Parasite infections were then monitored every 48 h by anaesthetizing fish and counting the total number of gyrodactylids over the first 17 days of infections; a timeline determined from our previous *G. turnbulli* infections (e.g. [37]).

To determine whether there was any immediate impact of noise exposure on *G. turnbulli* reproduction on the host, we infected $n = 10$ size-matched female guppies from the same mixed ornamental stock with 15 parasites each and exposed them to 24 h of noise as detailed above. Control fish ($n = 10$) were also infected but not exposed to noise. Over the 24 h time period, fish were removed and screened at two different time points (2 h and 24 h) to record parasite infrapopulation numbers.

Mortality was recorded for all treatments and any fish that survived parasite infection studies were treated with Levamisole (Norbrook®) according to Schelkle *et al.* [38]. Post-treatment fish were monitored for three weeks and no mortalities occurred during this time. Fish mortality only occurred during infections.

## 2.4. Statistical analysis

All statistical analyses were conducted using RStudio v. 2.1 [39] and final models were all selected based on the lowest Akaike's information criterion (AIC) value. Peak parasite burden is the maximum number

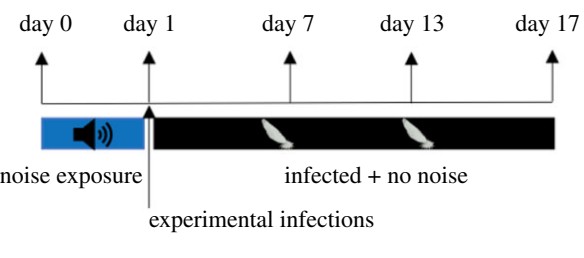

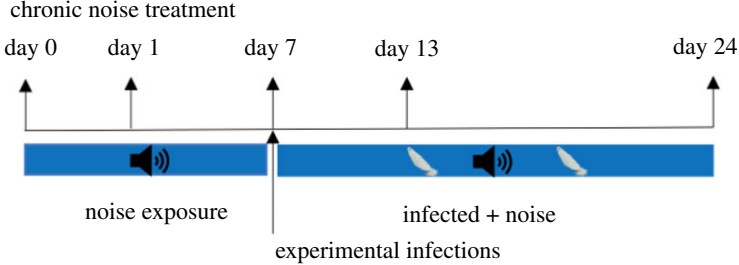

**Figure 3.** Timeline of when hosts were exposed to noise and experimental infection for both acute and chronic noise treatments.

of parasites at a given time point, defined here as peak day [40]. To quantify the total infection trajectory over the 17 days, we calculated the area under the curve (AUC) using the trapezoid rule [41]. To analyse peak parasite burden, peak day and AUC we used a generalized linear model (GLM) with a negative binomial error family and a log link function in the R MASS package. Explanatory variables for the GLM were treatment (no noise, acute noise, chronic noise) standard length and mortality day. All GLM error families were chosen based on the lowest dispersion parameter, theta [42].

A generalized linear mixed model (GLMM) with a negative binomial error family and log link function was used to analyse the intrinsic rate of parasite increase. A GLMM was used, as parasite data were recorded for each fish at different time points, and therefore to prevent pseudo-replication, Fish ID was treated as a random factor. Standard length and treatment (no noise, acute noise, chronic noise) were treated as explanatory variables. As experimental fish were placed in $n = 6$ (acute treatment) and $n = 7$ (chronic treatment) tanks, tank number was also treated as a fixed factor to rule out batch effect. For all models used in analysis, no batch effect was found for either noise exposure treatments ($p > 0.05$ for all models). Model refinement was conducted by removing standard length in the GLM and GLMM used to analyse AUC and intrinsic rates of parasite increase, as it was a non-significant explanatory variable (AUC: $Z = 0.72$, s.e. $= 0.01$, $p > 0.05$; intrinsic rate of parasite increase: $Z = 0.719$, s.e. $= 0.04$, $p > 0.05$).

For analysing the *in vivo* impact of 24 h noise exposure on parasite infrapopulations, a GLMM with a Poisson error family was used to analyse parasite count over time to prevent pseudo-replication as fish were screened at two different time points. The explanatory variable for this model being 'treatment' (i.e. noise exposure versus no-sound) and fish ID being a fixed factor. A further GLMM with a negative binomial error family and log link function was used to analyse parasite counts on hosts that survived till day 17 (days 13–17). Here, the explanatory variables were standard length and treatment. Finally, A GLM with Poisson error family and log link function was used to analyse death day, where the explanatory variable was treatment, as fish mortality only occurred if they were infected.

## 3. Results

Guppies exposed to acute noise and subsequently infected had significantly greater parasite burdens over time as measured through AUC compared with no noise controls (GLM: $Z = 0.08$, s.e. $= -4.14$, $p < 0.001$; figure 4$a$). Fish exposed to acute noise also had significantly higher peak parasite burdens compared with controls (GLM: $Z = -6.44$, s.e. $= 0.09$, $p < 0.001$; figure 4$b$). By contrast, guppies exposed to chronic noise had significantly reduced peak parasite burden and infection trajectories compared with controls (GLM, peak parasite burden: $Z = -8.4$, s.e. $= 0.07$, $p < 0.001$; AUC: $Z = -9.9$, s.e. $= 0.06$, $p < 0.001$). Fish exposed to chronic noise also showed a reduced intrinsic rate of parasite increase

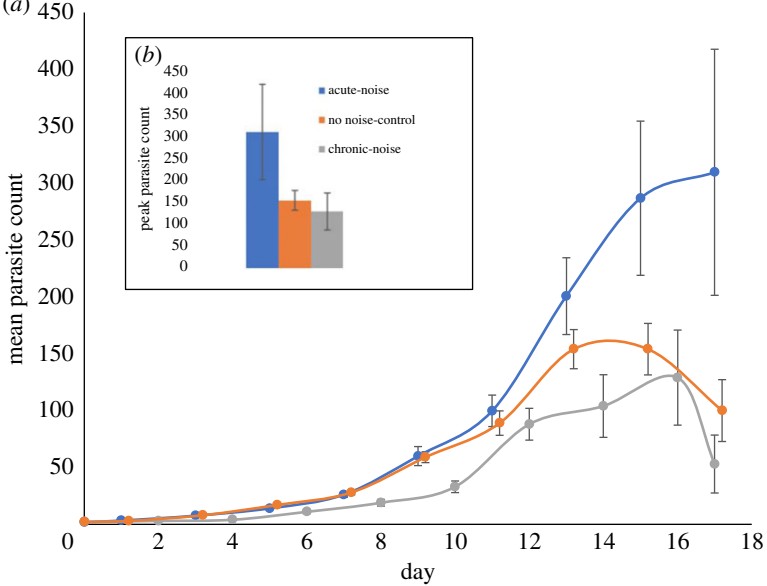

**Figure 4.** Mean count (*a*) and peak parasite burden (*b*) of *Gyrodactylus turnbulli* infections in guppies (*Poecilia reticulata*) exposed to either acute, chronic or no noise (controls). Standard error bars slightly transposed to one side to prevent overlap.

compared with control guppies (GLMM: $Z = -3.554$, s.e. $= 0.10219$, $p < 0.01$). When parasites were directly exposed *in vivo* to noise, the number of worms alive did not significantly change over the 24 h time period (GLMM: $Z = -0.43$, s.e. $= 0.06$, $p > 0.05$) suggesting no immediate impact of sound exposure on the parasites.

Day on which host mortality occurred was significantly associated with peak parasite count and AUC for both acute and chronic noise treatments (AUC: $Z = 52.23$, s.e. $= 0.007$, $p < 0.001$; peak count: $Z = 39.8$, s.e. $= 0.008$, $p < 0.001$) indicating that mortality of hosts influenced overall infection trajectories. Furthermore, when analysing parasite counts on only those hosts that survived until day 17 of the infection, no significant difference was seen between chronic noise treatments and controls (parasite count from day 13–17: GLMM: $Z = -1.4$, s.e. $= 0.88$, $p > 0.05$). This reflects the fact that guppies experiencing chronic noise treatment died significantly earlier than fish experiencing acute or no noise (chronic treatment average death day = 12, compared with average death day = 14 for acute and control fish, GLM: $Z = 3.08$, s.e. $= 0.03723$, $p < 0.01$).

## 4. Discussion

Anthropogenic noise pollution is now a recognized welfare concern, with international regulations (e.g. [43]) aiming to restrict potential detrimental health impacts. Regulations in the European Parliament, for example, have imposed restrictions on noise levels for motorized vehicles as well as introducing silencing systems, in recognition that noise can have wide-ranging health impacts. Here, we show fish experiencing acute noise suffered from increased disease susceptibility. By contrast, chronic noise exposure significantly reduced parasite burden, but fish were prone to earlier mortality. It is likely that acute noise caused a stress response without providing sufficient time for the immune system to respond before pathogenic challenge. Acute stress has been linked to increased lymphocyte trafficking and expressions of protein cytokines from leucocytes, whereas chronic stress is associated with reduced leucocyte function [44,45]. Chronic stress is generally accepted as detrimental to immunity and acute stress as potentially adaptive [44]. However, our understanding of how stressors impact the immune system is largely based on *in vitro* investigations (reviewed in [46]), although we are seeing an increase in studies on how stressors influence disease resistance. This is unsurprising considering the economic cost of disease for animal husbandry [16,22] and certainly for farmed aquatic species classic stressors, such as stocking density and water quality, are now known to significantly impact immune responses and disease resistance [47,48]. In addition to stressors associated with husbandry practices, those linked to environmental change are also being investigated in relation to disease resistance [49]. There is particular concern regarding such stressors that cross critical thresholds, termed 'planetary

boundaries' [50], that induce physiological stress leading to system dysfunctions that include increased disease susceptibility [51]. Noise pollution, however, that may be contributing to the breach of planetary boundaries has previously been neglected in terms of disease resistance. Therefore, *in vivo* experiments, combined with immunological expression studies, are needed to determine how noise has functional impacts on disease resistance.

Chronic noise exposure can activate the immune system, with gilthead sea bream (*Sparus aurata*), for example, showing significantly higher total oxidant status, lysozyme activity and antiprotease activity in response to 40 days of chronic aquaculture noise compared with no noise controls [18]. Chronic noise exposure in mice can cause immune alterations but this is dependent on strain type, with T-cell-dependent antibody production and *ex vivo* T-cell proliferation significantly reduced in C57Bl/6 but not BALB/c mice [52]. In comparison with a classic stressor (physical restraint) also applied to the C57Bl/6 mice, chronic noise had a greater impact on antibody production and immune cell proliferation (see [50]). In our study, chronic noise exposure was linked to significantly earlier host mortality and reduced pathogen burdens compared with fish from acute noise and control treatments, which strongly indicates that chronic noise reduces pathogen tolerance. We cannot exclude the fact that direct exposure to sound might also disrupt parasite development or cause them to actively move off the host. However, our *in vivo* investigations suggest that, at least over 24 h, noise exposure has no immediate impact on parasite infrapopulations. The only research showing that sound exposure can directly impact ectoparasites used ultrasonic waves that are of frequencies several orders of magnitude higher (e.g. [53]) than those used in the current study. Furthermore, at ultrasonic frequencies, sound only impacted ectoparasitic lice when they were within close range of the emitted sound (see [53]).

Animal food industries, including aquaculture, are projected to see a further rise in disease burden linked to increased stressors [16,54]. Here, for the first time, we reveal the detrimental impact of noise exposure on disease resistance and mortality. With animal husbandry focused on increasing output to meet human food chain demands, increased automation and machinery use is exposing animals to further noise [11,13]. We are aware that our study has isolated noise as an individual stressor under laboratory conditions and that animals face multiple stressors during routine husbandry. Future work must consider how noise pollution in conjunction with other common anthropogenic stressors, for instance enrichment use [55], transport [25] and manual handling [56], impact animal health. Currently, there are no effective treatments for many of the diseases that plague animal industries and the renewed emphasis on 'prevention rather than cure' means that now more than ever identifying key stressors associated with increased disease burden is an important goal towards developing sustainable preventive measures.

Ethics. All animal work was approved by Cardiff Universities Animal Ethics Committee and conducted under UK Home Office Licence (PPL 303424) following ARRIVE guidelines.

Data accessibility. Data is publicly available at Dryad [57] and can be accessed via the following: https://doi.org/10.5061/dryad.tmpg4f4v6.

Authors' contributions. N.M. and J.C. designed the study and drafted the manuscript. N.M. and L.H. conducted all experimental infections and associated data analyses. D.C. and S.G. collected and analysed the noise data. All authors commented on the final manuscript.

Competing interests. The authors declare no competing interests.

Funding. There is no funding to declare for this research.

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
