## [Reviewer comments · Royal Society Open Science]

Review History

RSOS-200172.R0 (Original submission)

Review form: Reviewer 1

Is the manuscript scientifically sound in its present form?

Yes

Are the interpretations and conclusions justified by the results?

Yes

Is the language acceptable?

Yes

Do you have any ethical concerns with this paper?

No

Have you any concerns about statistical analyses in this paper?

No

Recommendation?

Accept with minor revision (please list in comments)

Comments to the Author(s)

This is a very well written paper on an important topic in fish husbandry that justifies publication. Minor comments:

Figure 1 legend - maintain a space between 1 l as it looks like 11.

Results text, first sentence - The first figure referred to relates to peak parasite burden which is Figure 2B, therefore some adjustment is needed in the figure and the second mention of the figure in the next sentence.

A very interesting discussion. Are there any studies on the response of ectoparasites to sound waves in water or vibration?

Review form: Reviewer 2

Is the manuscript scientifically sound in its present form?

No

Are the interpretations and conclusions justified by the results?

Yes

Is the language acceptable?

Yes

Do you have any ethical concerns with this paper?

No

Have you any concerns about statistical analyses in this paper?

No

Recommendation?

Accept with minor revision (please list in comments)

Comments to the Author(s)

This study provides a topical approach in understanding the effects of acoustic disturbance on fish health. The effects of noise on tertiary responses relating to fish health is largely understudied, and this work would contribute to improving the welfare of captive fish populations, and benefit the field as a whole. I'm generally positive about the study. However, I have a number of key reservations which should be addressed before being considered for publication.

The lack of formal sound analysis and presentation of acoustic data (treatments) in the manuscript is a concern. It's vital to provide full details of the stressor applied throughout the experiment. At the very least, I would like to see sound pressure levels characterised and presented for the different playback treatments. These sound levels should be calculated over the likely hearing range of the study species; presentation of mean sound levels over a wide bandwidth is misleading. Without such information it's unknown whether the levels presented were ecologically relevant, as well as precluding replication of the study.

White noise has a flat power spectrum, however playback in small tanks may drastically alter this through reflections and reverberations. As such, it would be best practice to show how the sound was altered. Further, I would like to see how the sound treatments compared to the background noise of aquaria and/or holding tank acoustic conditions. In addition, sound pressure represents

only one element of the sound field, and mention of particle motion should be included. Granted, the size and expense of accelerometers may prevent their use, but it's highly likely that guppies will be responding to particle motion than pressure. Reference should be made to particle motion even if it wasn't able to be measured at this time.

The manuscript is lacking in detail which made it difficult to follow in places. More clarity is needed on the exact noise exposure regime. Line 105, states chronic noise consists of 7 days of exposure, but L144-L147 counters this stating chronic noise lasted a total duration of 24 days. Perhaps an additional figure showing the experimental design would be beneficial, detailing when the noise exposure started for each treatment, when parasites were added, and how the sampling regime was undertaken. The methods need more detail describing exactly how all response measures were collected. Mortality as a metric first appears in the results, so comes out of the blue after reading the methods.

The statistical analysis section is missing detail on if and how model assumptions were checked in line with use of GLMs and GLMMs. It would be useful to present the full model outputs of each analysis in tables.

The use of noise pollution in the title and throughout the manuscript should be amended. It's difficult to link it to noise pollution in general as playback tracks all used white noise. As such, I think it more appropriate to frame the work and stimulus as an acoustic disturbance or, simply, noise exposure.

Additional comments:

L58: Range of tertiary responses have been shown with regards to fish behaviour. Suggest amend to reflect that very limited work on disease resistance in particular.

L100-101: Want to know sizes (standard length and mass) of fish in each treatment group, and tests to show all the same, with standard error presented.

L118: UW-30s have a frequency response of 100-10,000Hz.

L121-122: Clarification needed on what exactly this means. Was the speaker on, but playing nothing, or completely switched off and disconnected from the battery? This detail only appears in the figure 1 legend.

L180-181: Chi-squared test used to analyse host mortality, but then host mortality data in the statistics was analysed with a GLM (L197).

L189-197: Appears that references to figures at the end of the within-text stats is in the wrong order. I.e. Area Under the Curve is made in reference to figure 2B, the bar plot of peak parasite count.

Figure 2: Needs mention of control groups in figure legend.

L211: Need for references showing acoustic disturbance can act as a stressor (primary, secondary mechanisms).

L244-245: Number of papers have shown detrimental impact on at least mortality (e.g. Nedelec et al., 2015, Scientific Reports, 4, 5891). Suggest revision of this statement to include caveats, and directly tie in to the results found in this study.

Review form: Reviewer 3

Is the manuscript scientifically sound in its present form?

Yes

Are the interpretations and conclusions justified by the results?

No

Is the language acceptable?

Yes

Do you have any ethical concerns with this paper?

No

Have you any concerns about statistical analyses in this paper?

No

Recommendation?

Accept with minor revision (please list in comments)

Comments to the Author(s)

This is an interesting manuscript on a timely topic that is clearly presented. I think that the findings are important and worthy of publication in Royal Society Open Science. In my view, the authors need to clarify some details of the research protocols and (importantly) reconsider one area of their conclusions.

The authors have investigated the impact of underwater noise on parasitism in guppies. They report that acute exposure to noise before infection resulted in the development of significantly greater parasite burdens relative to controls. In contrast, fish in a treatment exposed to chronic noise throughout the experiment developed lower parasite burdens but died earlier. I found the first result especially interesting: it is striking to see that a transient exposure to 24hrs noise resulted in higher burdens developing around 2 weeks later.

Comment on the conclusions:

The most important concern I have about the manuscript is the interpretation of the results from the chronic sound treatment. Fish in this chronic sound treatment developed lower parasite burdens than fish in the control treatment, yet the fish died earlier. The authors put forward an interesting hypothesis that the chronic sound treatment may upregulate fish immune responses, that these immune responses help to reduce parasite burden, but that the responses themselves are costly and these immune costs lead to shortened lifespan. This hypothesis is worth reporting, yet it is not the only hypothesis consistent with the data. I think it is important for the authors to consider alternative hypotheses in the text, to acknowledge more clearly that this hypothesis is rather speculative (in the absence of any data about immune system activity etc), and to reconsider (reduce?) the prominence of this hypothesis in the abstract.

The reason for my concerns is based on my understanding of the experimental design. In this chronic noise treatment fish were exposed to noise for 6 days, then infected with 2 parasites on the 7th day (during which time noise exposure continued), then noise exposure continued for a further 17 days. This treatment is compared to the control treatment (where fish were not exposed to noise), and the acute treatment (where fish were exposed to noise for 24 hours, the noise was stopped, and the fish were then infected with parasites).

For me, the key difference between the chronic and the acute treatments (which the authors do not mention in the manuscript) is that in the chronic treatment the parasites were exposed to the

noise treatment (in addition to the fish), whereas in the acute treatment the parasites were never exposed to noise. This means that the authors cannot exclude the possibility that the low parasite burden in the chronic noise treatment might be caused by the direct impact of noise on the parasite (rather than the parasites being indirectly affected via an immune response putatively upregulated in the host in response to chronic noise – as the authors suggest).

Clearly there is no evidence that the parasites are indeed directly affected by chronic noise (and if they are, this could be a very important and interesting observation – of potential fundamental and applied significance). However, it strikes me that this direct effect of sound on the parasite is no less likely than the hypothesis the authors put forward (indeed impacts of noise on other invertebrate animals are known from other studies).

To summarise, the comparison between the acute and control treatment is relatively straightforward to interpret (as the authors have done), because only the fish experienced the noise stress. Whereas in the chronic noise treatment both the parasite and the fish experienced the noise, therefore the authors need to consider how effects of noise on both organisms could have driven the observed results.

Other points for clarification etc:

Methods:

The methods section is perfectly well written, but I found the precise details of the timeline of the experiment slightly difficult to get straight in my head when I read the methods. Did the chronic exposure start 6 days before the acute exposure, then both groups infected on the same day? A timeline diagram in supplementary materials would help here, combined with some changes in the text.

Is there any difference in mean age of the fish at the time of infection driven by the timing of the sound treatments – if so, some comments would be useful.

Can the authors clarify why the fish were treated to remove infection on day 17? I think the fact that infection and sound treatment stopped on day 17 potentially complicates the interpretation of subsequent mortality observations.

Results:

The presentation of the infection data is very clear (and interesting). However, the results for differences in lifespan between treatments are covered in a single sentence with no figure. Would the authors consider plotting a survival curve for the three treatments, even if this ends up as a supplementary figure in the final paper? I would guess that all the fish died during the experiment, but this is not stated explicitly – could the authors clarify this. If all the fish did not die, survival analysis would be more appropriate than the GLM reported).

Could the authors comment on (and/or include appropriate analysis) what influence selective disappearance of individuals could potentially have on the parasite infection metrics? I note that average death date is around day 12 / day 14 which is about the same time that compelling treatment differences in infection metrics arise (figure 2). If deaths were non-random with respect to infection status then this might influence interpretation of the results.

Review form: Reviewer 4

Is the manuscript scientifically sound in its present form?

Yes

Are the interpretations and conclusions justified by the results?

Yes

Is the language acceptable?

Yes

Do you have any ethical concerns with this paper?

No

Have you any concerns about statistical analyses in this paper?

No

Recommendation?

Accept with minor revision (please list in comments)

Comments to the Author(s)

This paper describes an elegantly designed experiment to test how noise pollution affects parasite burden (and by inference, immune response) in the guppy-gyrodactylid model system. The results are important – acute noise pollution impacts parasite burden significantly, and chronic noise pollution leads to a decrease in host lifespan. This paper will be an important reference for future work.

Some specific comments below:

L19-22- the sentence about the importance of parasites could do with splitting into two, so that ecosystem biodiversity and the animal trade are separated concepts

L69-73 – the importance of parasites is explained here, but not in the context of noise, and how much of an exacerbating factor noise might be

L76-77 – add what was found in these studies i.e. “(e.g. nitrate enrichment reduced disease 24....)”

L85 again, it isn’t outlined how much of a risk noise pollution is to global freshwater species with specific examples

Figure 1 A doesn’t add much to the verbal description of the experiment at the moment – could this be redesigned to include all three treatments more clearly, as well as the split into infection vs no infection, with numbers on the schematic?

L183 data is clearly presented in the Dryad file – for full reproducibility marks, the authors should consider also publishing their R code.

(What an utter joy it was to review a paper where the figures appear in the Results, instead of having to scroll to the end).

L211. “fish were prone to earlier mortality” could be phrased more decisively, since there was a significant difference in death day.

In the analysis, peak parasite burden and peak day are not independent, since it would take longer for parasite burden to get to a larger number. Would it be worth removing this extra factor from the analysis, or making it clearer if it is indeed independent?

In the intro, lines 56-59, responses to noise are described – perhaps the Discussion of findings here could refer back to this.

What might the impacts of noise pollution be on the parasites?

Decision letter (RSOS-200172.R0)

Dear Mr Masud

On behalf of the Editors, I am pleased to inform you that your Manuscript RSOS-200172 entitled "Noise pollution: acute noise exposure increases susceptibility to disease and chronic exposure reduces host survival" has been accepted for publication in Royal Society Open Science subject to minor revision in accordance with the referee suggestions. Please find the referees' comments at the end of this email.

The reviewers and handling editors have recommended publication, but also suggest some minor revisions to your manuscript. Therefore, I invite you to respond to the comments and revise your manuscript.

- Ethics statement

- Data accessibility

If you wish to submit your supporting data or code to Dryad (<http://datadryad.org/>), or modify your current submission to dryad, please use the following link:
<http://datadryad.org/submit?journalID=RSOS&manu=RSOS-200172>

- Competing interests

- Authors' contributions

- Acknowledgements

- Funding statement

Because the schedule for publication is very tight, it is a condition of publication that you submit the revised version of your manuscript before 27-May-2020. Please note that the revision deadline will expire at 00.00am on this date. If you do not think you will be able to meet this date please let me know immediately.

- 1) A text file of the manuscript (tex, txt, rtf, docx or doc), references, tables (including captions) and figure captions. Do not upload a PDF as your "Main Document";

- 2) A separate electronic file of each figure (EPS or print-quality PDF preferred (either format should be produced directly from original creation package), or original software format);
- 3) Included a 100 word media summary of your paper when requested at submission. Please ensure you have entered correct contact details (email, institution and telephone) in your user account;
- 4) Included the raw data to support the claims made in your paper. You can either include your data as electronic supplementary material or upload to a repository and include the relevant doi within your manuscript. Make sure it is clear in your data accessibility statement how the data can be accessed;
- 5) All supplementary materials accompanying an accepted article will be treated as in their final form. Note that the Royal Society will neither edit nor typeset supplementary material and it will be hosted as provided. Please ensure that the supplementary material includes the paper details where possible (authors, article title, journal name).

If your manuscript is newly submitted and subsequently accepted for publication, you will be asked to pay the article processing charge, unless you request a waiver and this is approved by Royal Society Publishing. You can find out more about the charges at <https://royalsocietypublishing.org/rsos/charges>. Should you have any queries, please contact openscience@royalsociety.org.

on behalf of Dr Peter Keller (Associate Editor) and Pete Smith (Subject Editor)
openscience@royalsociety.org

Associate Editor Comments to Author (Dr Peter Keller):
Comments to the Author:

Three experts have reviewed your manuscript and found the study to be interesting, topical, and important. However, all reviewers provided suggestions for minor revisions. In particular, Reviewer 2 requested the inclusion of formal acoustic analysis and Reviewer 3 questioned aspects of the interpretation of your results. All reviewers' concerns would need to be addressed before the manuscript could be accepted for publication.

Reviewer comments to Author:

Reviewer: 1

Comments to the Author(s)

This is a very well written paper on an important topic in fish husbandry that justifies publication. Minor comments:

Figure 1 legend - maintain a space between 1 l as it looks like 11.

Results text, first sentence - The first figure referred to relates to peak parasite burden which is Figure 2B, therefore some adjustment is needed in the figure and the second mention of the figure in the next sentence.

A very interesting discussion. Are there any studies on the response of ectoparasites to sound waves in water or vibration?

Reviewer: 2

Comments to the Author(s)

This study provides a topical approach in understanding the effects of acoustic disturbance on fish health. The effects of noise on tertiary responses relating to fish health is largely understudied, and this work would contribute to improving the welfare of captive fish populations, and benefit the field as a whole. I'm generally positive about the study. However, I have a number of key reservations which should be addressed before being considered for publication.

The lack of formal sound analysis and presentation of acoustic data (treatments) in the manuscript is a concern. It's vital to provide full details of the stressor applied throughout the experiment. At the very least, I would like to see sound pressure levels characterised and presented for the different playback treatments. These sound levels should be calculated over the likely hearing range of the study species; presentation of mean sound levels over a wide bandwidth is misleading. Without such information it's unknown whether the levels presented were ecologically relevant, as well as precluding replication of the study.

White noise has a flat power spectrum, however playback in small tanks may drastically alter this through reflections and reverberations. As such, it would be best practice to show how the sound was altered. Further, I would like to see how the sound treatments compared to the background noise of aquaria and/or holding tank acoustic conditions. In addition, sound pressure represents only one element of the sound field, and mention of particle motion should be included. Granted, the size and expense of accelerometers may prevent their use, but it's highly likely that guppies will be responding to particle motion than pressure. Reference should be made to particle motion even if it wasn't able to be measured at this time.

The manuscript is lacking in detail which made it difficult to follow in places. More clarity is needed on the exact noise exposure regime. Line 105, states chronic noise consists of 7 days of exposure, but L144-L147 counters this stating chronic noise lasted a total duration of 24 days. Perhaps an additional figure showing the experimental design would be beneficial, detailing when the noise exposure started for each treatment, when parasites were added, and how the sampling regime was undertaken. The methods need more detail describing exactly how all response measures were collected. Mortality as a metric first appears in the results, so comes out of the blue after reading the methods.

The statistical analysis section is missing detail on if and how model assumptions were checked in line with use of GLMs and GLMMs. It would be useful to present the full model outputs of each analysis in tables.

The use of noise pollution in the title and throughout the manuscript should be amended. It's difficult to link it to noise pollution in general as playback tracks all used white noise. As such, I think it more appropriate to frame the work and stimulus as an acoustic disturbance or, simply, noise exposure.

Additional comments:

L58: Range of tertiary responses have been shown with regards to fish behaviour. Suggest amend to reflect that very limited work on disease resistance in particular.

L100-101: Want to know sizes (standard length and mass) of fish in each treatment group, and tests to show all the same, with standard error presented.

L118: UW-30s have a frequency response of 100-10,000Hz.

L121-122: Clarification needed on what exactly this means. Was the speaker on, but playing nothing, or completely switched off and disconnected from the battery? This detail only appears in the figure 1 legend.

L180-181: Chi-squared test used to analyse host mortality, but then host mortality data in the statistics was analysed with a GLM (L197).

L189-197: Appears that references to figures at the end of the within-text stats is in the wrong order. I.e. Area Under the Curve is made in reference to figure 2B, the bar plot of peak parasite count.

Figure 2: Needs mention of control groups in figure legend.

L211: Need for references showing acoustic disturbance can act as a stressor (primary, secondary mechanisms).

L244-245: Number of papers have shown detrimental impact on at least mortality (e.g. Nedelec et al., 2015, *Scientific Reports*, 4, 5891). Suggest revision of this statement to include caveats, and directly tie in to the results found in this study.

Reviewer: 3

Comments to the Author(s)

This is an interesting manuscript on a timely topic that is clearly presented. I think that the findings are important and worthy of publication in *Royal Society Open Science*. In my view, the authors need to clarify some details of the research protocols and (importantly) reconsider one area of their conclusions.

The authors have investigated the impact of underwater noise on parasitism in guppies. They report that acute exposure to noise before infection resulted in the development of significantly greater parasite burdens relative to controls. In contrast, fish in a treatment exposed to chronic noise throughout the experiment developed lower parasite burdens but died earlier. I found the first result especially interesting: it is striking to see that a transient exposure to 24hrs noise resulted in higher burdens developing around 2 weeks later.

Comment on the conclusions:

The most important concern I have about the manuscript is the interpretation of the results from the chronic sound treatment. Fish in this chronic sound treatment developed lower parasite burdens than fish in the control treatment, yet the fish died earlier. The authors put forward an

interesting hypothesis that the chronic sound treatment may upregulate fish immune responses, that these immune responses help to reduce parasite burden, but that the responses themselves are costly and these immune costs lead to shortened lifespan. This hypothesis is worth reporting, yet it is not the only hypothesis consistent with the data. I think it is important for the authors to consider alternative hypotheses in the text, to acknowledge more clearly that this hypothesis is rather speculative (in the absence of any data about immune system activity etc), and to reconsider (reduce?) the prominence of this hypothesis in the abstract.

The reason for my concerns is based on my understanding of the experimental design. In this chronic noise treatment fish were exposed to noise for 6 days, then infected with 2 parasites on the 7th day (during which time noise exposure continued), then noise exposure continued for a further 17 days. This treatment is compared to the control treatment (where fish were not exposed to noise), and the acute treatment (where fish were exposed to noise for 24 hours, the noise was stopped, and the fish were then infected with parasites).

For me, the key difference between the chronic and the acute treatments (which the authors do not mention in the manuscript) is that in the chronic treatment the parasites were exposed to the noise treatment (in addition to the fish), whereas in the acute treatment the parasites were never exposed to noise. This means that the authors cannot exclude the possibility that the low parasite burden in the chronic noise treatment might be caused by the direct impact of noise on the parasite (rather than the parasites being indirectly affected via an immune response putatively upregulated in the host in response to chronic noise – as the authors suggest).

Clearly there is no evidence that the parasites are indeed directly affected by chronic noise (and if they are, this could be a very important and interesting observation – of potential fundamental and applied significance). However, it strikes me that this direct effect of sound on the parasite is no less likely than the hypothesis the authors put forward (indeed impacts of noise on other invertebrate animals are known from other studies).

To summarise, the comparison between the acute and control treatment is relatively straightforward to interpret (as the authors have done), because only the fish experienced the noise stress. Whereas in the chronic noise treatment both the parasite and the fish experienced the noise, therefore the authors need to consider how effects of noise on both organisms could have driven the observed results.

Other points for clarification etc:

Methods:

The methods section is perfectly well written, but I found the precise details of the timeline of the experiment slightly difficult to get straight in my head when I read the methods. Did the chronic exposure start 6 days before the acute exposure, then both groups infected on the same day? A timeline diagram in supplementary materials would help here, combined with some changes in the text.

Is there any difference in mean age of the fish at the time of infection driven by the timing of the sound treatments – if so, some comments would be useful.

Can the authors clarify why the fish were treated to remove infection on day 17? I think the fact that infection and sound treatment stopped on day 17 potentially complicates the interpretation of subsequent mortality observations.

Results:

The presentation of the infection data is very clear (and interesting). However, the results for differences in lifespan between treatments are covered in a single sentence with no figure. Would

the authors consider plotting a survival curve for the three treatments, even if this ends up as a supplementary figure in the final paper? I would guess that all the fish died during the experiment, but this is not stated explicitly – could the authors clarify this. If all the fish did not die, survival analysis would be more appropriate than the GLM reported).

Could the authors comment on (and/or include appropriate analysis) what influence selective disappearance of individuals could potentially have on the parasite infection metrics? I note that average death date is around day 12 / day 14 which is about the same time that compelling treatment differences in infection metrics arise (figure 2). If deaths were non-random with respect to infection status then this might influence interpretation of the results.

Author's Response to Decision Letter for (RSOS-200172.R0)

See Appendix A.

RSOS-200172.R1 (Revision)

Review form: Reviewer 2

Is the manuscript scientifically sound in its present form?

Yes

Are the interpretations and conclusions justified by the results?

Yes

Is the language acceptable?

Yes

Do you have any ethical concerns with this paper?

No

Have you any concerns about statistical analyses in this paper?

No

Recommendation?

Accept with minor revision (please list in comments)

Comments to the Author(s)

The authors have done a great job in responding to the comments raised. This has resulted in a much improved manuscript that will be a valuable addition to the field.

There is, however, one additional point which should be addressed before publication. The authors have now clarified that the speakers in the control trials were turned off. This creates a potential confound, whereby a magnetic field is present in the treatment trials but not the control trials. I suggest a single sentence is added in the discussion or methods to highlight this.

Review form: Reviewer 3

Is the manuscript scientifically sound in its present form?

Yes

Are the interpretations and conclusions justified by the results?

No

Is the language acceptable?

Yes

Do you have any ethical concerns with this paper?

No

Have you any concerns about statistical analyses in this paper?

Yes

Recommendation?

Accept with minor revision (please list in comments)

Comments to the Author(s)

I am pleased to see the changes that the authors have made since the previous round of review – I think this has resulted in an improved manuscript, which in my opinion is now close to being ready for acceptance. Almost all my comments in this review concern the difference that the authors describe between the control treatment and the chronic sound treatment. My comments outline that there are probably plausible alternative hypotheses to the hypothesis the authors favour in their discussion and abstract.

I must admit to being concerned that the authors have striven to retain statements in the paper that unambiguously claim to demonstrate that chronic sound stimulates immune responses and that investment in these immune responses shortens fish lifespan. I don't think there is any strong evidence in the paper that clearly demonstrates that this is true. To address this point the authors need to do little more than change one line in the abstract and one line in the discussion. With or without this claim, the authors still have a very interesting paper, and the fact that the other sound treatment (acute noise before parasite infection) has such an impressive effect on infection load makes the paper exciting even without the claim that the authors have retained in the manuscript regarding immune upregulation and life history costs.

To address this point following the previous review, the authors have included new experimental data and made alterations to the discussion.

The new data are from an experiment delivering an acute 24 hour exposure of sound to parasites on hosts; the data verify that parasites do not drop off or die during this 24 hour period.

However, the new data do not provide a comparator treatment for the chronic sound treatment in the main paper where fish¶sites were exposed to sound for seven days. I find the argument that this new data set demonstrates that there is "no direct impact of sound on the parasites" (line 231) unconvincing because in this new experiment the parasite count was only carried out during the 24 hour time period when the fish were receiving the sound treatment (with no monitoring thereafter). This new experiment provides interesting data for the authors to include and the extra information adds to the paper. However, the result cannot be used to argue that (compared to controls) the difference in parasite numbers in the chronic exposure treatment (which lasted 14 days) can only be due to physiological changes in the fish (and not potential direct effects on the parasites).

I am content that the authors have added some well explained text to the discussion to mention the alternative hypothesis that there could be direct effects of chronic sound on the parasites. The

authors now outline the possibilities that chronic sound directly impacts parasites to cause lower burdens vs the possibility that chronic sound activates the fish immune system to cause chronic burdens. However, these text changes then lead the authors to write a new concluding sentence in this section of the discussion (lines 295-298) that “While exposure to noise has been shown to impact fish mortality this is the first study to demonstrate how increased disease resistance linked to chronic noise exposure reduces survival”. This is simply not true – the authors have not demonstrated that disease resistance increases in fish that receive chronic noise exposure (although lower parasite burdens might be consistent with this), furthermore they have certainly not demonstrated that this putative change in resistance is the factor that reduces fish survival (other direct effects of sound on the fish are possible).

I also think it is disappointing that there has been no change in wording to the abstract in the new manuscript in relation to this issue. The current wording says that differences between the chronic sound and control treatments “demonstrating a potential functional trade-off between improved parasite resistance and shorter lifespan” doesn’t really acknowledge the uncertainty or speculative nature of this claim.

The authors have made a brief response to my previous review comment about whether non-random mortality might cause selective disappearance of fish from the data set that might influence the results. The key issue I was raising was that if those fish that have the highest parasite burdens are most likely to die in the chronic infection treatment, then the fact that mean parasite burden is lower in this treatment could possibly be driven by the fact that the highly infected fish have died, thereby causing a low mean parasite burden at later timepoints. This issue is especially important because the chronic treatment has the highest/earliest mortality rate, thus the potential effect of selective disappearance would be greatest in this treatment. The authors include new analysis which makes me more concerned about this issue, not less concerned. The new analysis that the authors present indicates that mortality probably is being driven by infection - it appears that those fish that die earliest are those that have highest parasite burdens. This reinforces an additional potential alternative explanation for the low parasite burdens in the chronic treatment - now there are 3: (i) chronic sound upregulates an immune response that protects the fish from infection (ii) chronic sound directly negatively affects the parasite and (iii) chronic sound lowers fish tolerance of infection pathology resulting in fish with high burdens dying early.

I would encourage the authors to address this issue briefly in the results section, with appropriate modifications to the discussion section where necessary. The authors could do some of the following things (or there may be other approaches):

1. Perhaps the best way to address selective disappearance in data sets like this is to include an additional variable for ‘age at death’ in the statistical model, which then leaves other variables in the model to assess effects on parasite metrics once selective disappearance has been accounted for.
2. The authors might also present the data for the days on which the fish died, rather than just the mean death days (I suggested they add a survival curve to the manuscript in my previous review). This might allow them to argue that (perhaps) substantial differences in mean parasite characteristics emerge in (old)fig3 before any mortality has occurred.
3. The authors could reanalyse data including only those fish that survive all the way through to (eg) day 15, if the difference between the chronic and control treatments still remains then then authors could conclude that selective disappearance isn’t important/the-only-factor driving the difference seen.

It may be that these analyses mean that the authors can conclude that selection disappearance is not important, which would simplify any changes required.

Other comments:

Lines 68-70 – introduction. In reference to Wysocki et al. it seems odd to phrase this sentence that this is the only study to ‘demonstrate an unambiguous effect’, when the sentence goes on to say that fish in this study were ‘unaffected by chronic 8-month noise exposure’. It is not clear to me what the “unambiguous effect” is.

On the new figure 3 it would be helpful, in the top panel, if the word ‘noise exposure’ did not spread over into the black area of the figure which represents ‘no noise’.

Line 181. Change ‘Controls fish’ to ‘Control fish’.

Line 196. Remove apostrophe in GLM’s.

Line 199 onwards. I may have misunderstood the analysis for this GLMM, but was time/day not also included as an explanatory variable in this test? I assume it was, due to the reference to repeated measures on fish at different time points. If time/day was a variable in the model this should be added to the text. However, in addition, could the authors clarify how they accounted for the fact that the effect of day is very non-linear (as shown in Fig3).

Line 209. Correct ‘explanatory variables’ to ‘explanatory variable’.

Line 214. Should fish ID be described as a random factor rather than a fixed factor?

Line 228. GLMM result – standardise the number of decimal places as with other results.

Line 238. GLM result – standardise the number of decimal places as with other results.

Line 243. The graph of parasite count through time should be figure 4, not figure 3 (references to this figure in the text also need changing).

Review form: Reviewer 4

Is the manuscript scientifically sound in its present form?

Yes

Are the interpretations and conclusions justified by the results?

Yes

Is the language acceptable?

Yes

Do you have any ethical concerns with this paper?

No

Have you any concerns about statistical analyses in this paper?

No

Recommendation?

Accept with minor revision (please list in comments)

Comments to the Author(s)

The extra clarification and in particular the extra experiment to investigate the effect of noise on the parasites is a really great addition to this paper.

There are a couple of minor issues the authors may wish to change:

Fig 3. There is an extra space in the legend

Within the figure, there is some white over some of the "Day" labels so they read as "Dav".

L181. Control fish (not controls fish)

L210. needs a comma for clarity.

One extra thing that could be clarified - in the response to reviewers letter, the authors say that a combined effect of sound and infection likely affect mortality - but surely this interaction term has been tested in the models? This needs clarifying I think.

Finally, I would consider adding a sentence explaining that the infections were removed after day 17 for ethical reasons.

Decision letter (RSOS-200172.R1)

Dear Mr Masud

On behalf of the Editors, we are pleased to inform you that your Manuscript RSOS-200172.R1 "Noise pollution: acute noise exposure increases susceptibility to disease and chronic exposure reduces host survival" has been accepted for publication in Royal Society Open Science subject to minor revision in accordance with the referees' reports. Please find the referees' comments along with any feedback from the Editors below my signature.

Please submit your revised manuscript and required files (see below) no later than 7 days from today's (ie 14-Aug-2020) date. Note: the ScholarOne system will 'lock' if submission of the revision is attempted 7 or more days after the deadline. If you do not think you will be able to meet this deadline please contact the editorial office immediately.

on behalf of Dr Peter Keller (Associate Editor) and Pete Smith (Subject Editor)
 openscience@royalsociety.org

Associate Editor Comments to Author (Dr Peter Keller):

All three reviewers recommend accepting the revised manuscript with further minor revisions. The comments made by Reviewer 3 in fact raise significant issues, and for that reason I would most likely send the next revision back to that reviewer for a final check.

Reviewer comments to Author:

Reviewer: 2

Comments to the Author(s)

The authors have done a great job in responding to the comments raised. This has resulted in a much improved manuscript that will be a valuable addition to the field.

There is, however, one additional point which should be addressed before publication. The authors have now clarified that the speakers in the control trials were turned off. This creates a potential confound, whereby a magnetic field is present in the treatment trials but not the control trials. I suggest a single sentence is added in the discussion or methods to highlight this.

Reviewer: 3

Comments to the Author(s)

I am pleased to see the changes that the authors have made since the previous round of review – I think this has resulted in an improved manuscript, which in my opinion is now close to being ready for acceptance. Almost all my comments in this review concern the difference that the authors describe between the control treatment and the chronic sound treatment. My comments outline that there are probably plausible alternative hypotheses to the hypothesis the authors favour in their discussion and abstract.

I must admit to being concerned that the authors have striven to retain statements in the paper that unambiguously claim to demonstrate that chronic sound stimulates immune responses and that investment in these immune responses shortens fish lifespan. I don't think there is any strong evidence in the paper that clearly demonstrates that this is true. To address this point the authors need to do little more than change one line in the abstract and one line in the discussion. With or without this claim, the authors still have a very interesting paper, and the fact that the other sound treatment (acute noise before parasite infection) has such an impressive effect on infection load makes the paper exciting even without the claim that the authors have retained in the manuscript regarding immune upregulation and life history costs.

To address this point following the previous review, the authors have included new experimental data and made alterations to the discussion.

The new data are from an experiment delivering an acute 24 hour exposure of sound to parasites on hosts; the data verify that parasites do not drop off or die during this 24 hour period.

However, the new data do not provide a comparator treatment for the chronic sound treatment in the main paper where fish¶sites were exposed to sound for seven days. I find the argument that this new data set demonstrates that there is “no direct impact of sound on the parasites” (line 231) unconvincing because in this new experiment the parasite count was only carried out during the 24 hour time period when the fish were receiving the sound treatment (with no monitoring thereafter). This new experiment provides interesting data for the authors to include and the extra information adds to the paper. However, the result cannot be used to argue that (compared to controls) the difference in parasite numbers in the chronic exposure treatment (which lasted 14

days) can only be due to physiological changes in the fish (and not potential direct effects on the parasites).

I am content that the authors have added some well explained text to the discussion to mention the alternative hypothesis that there could be direct effects of chronic sound on the parasites. The authors now outline the possibilities that chronic sound directly impacts parasites to cause lower burdens vs the possibility that chronic sound activates the fish immune system to cause chronic burdens. However, these text changes then lead the authors to write a new concluding sentence in this section of the discussion (lines 295-298) that “While exposure to noise has been shown to impact fish mortality this is the first study to demonstrate how increased disease resistance linked to chronic noise exposure reduces survival”. This is simply not true – the authors have not demonstrated that disease resistance increases in fish that receive chronic noise exposure (although lower parasite burdens might be consistent with this), furthermore they have certainly not demonstrated that this putative change in resistance is the factor that reduces fish survival (other direct effects of sound on the fish are possible).

I also think it is disappointing that there has been no change in wording to the abstract in the new manuscript in relation to this issue. The current wording says that differences between the chronic sound and control treatments “demonstrating a potential functional trade-off between improved parasite resistance and shorter lifespan” doesn’t really acknowledge the uncertainty or speculative nature of this claim.

The authors have made a brief response to my previous review comment about whether non-random mortality might cause selective disappearance of fish from the data set that might influence the results. The key issue I was raising was that if those fish that have the highest parasite burdens are most likely to die in the chronic infection treatment, then the fact that mean parasite burden is lower in this treatment could possibly be driven by the fact that the highly infected fish have died, thereby causing a low mean parasite burden at later timepoints. This issue is especially important because the chronic treatment has the highest/earliest mortality rate, thus the potential effect of selective disappearance would be greatest in this treatment. The authors include new analysis which makes me more concerned about this issue, not less concerned. The new analysis that the authors present indicates that mortality probably is being driven by infection - it appears that those fish that die earliest are those that have highest parasite burdens. This reinforces an additional potential alternative explanation for the low parasite burdens in the chronic treatment - now there are 3: (i) chronic sound upregulates an immune response that protects the fish from infection (ii) chronic sound directly negatively affects the parasite and (iii) chronic sound lowers fish tolerance of infection pathology resulting in fish with high burdens dying early.

I would encourage the authors to address this issue briefly in the results section, with appropriate modifications to the discussion session where necessary. The authors could do some of the following things (or there may be other approaches):

1. Perhaps the best way to address selective disappearance in data sets like this is to include an additional variable for ‘age at death’ in the statistical model, which then leaves other variables in the model to assess effects on parasite metrics once selective disappearance has been accounted for.
2. The authors might also present the data for the days on which the fish died, rather than just the mean death days (I suggested they add a survival curve to the manuscript in my previous review). This might allow them to argue that (perhaps) substantial differences in mean parasite characteristics emerge in (old)fig3 before any mortality has occurred.
3. The authors could reanalyse data including only those fish that survive all the way through to (eg) day 15, if the difference between the chronic and control treatments still remains then then

authors could conclude that selective disappearance isn't important/the-only-factor driving the difference seen.

It may be that these analyses mean that the authors can conclude that selection disappearance is not important, which would simplify any changes required.

Other comments:

Lines 68-70 – introduction. In reference to Wysocki et al. it seems odd to phrase this sentence that this is the only study to 'demonstrate an unambiguous effect', when the sentence goes on to say that fish in this study were 'unaffected by chronic 8-month noise exposure'. It is not clear to me what the "unambiguous effect" is.

On the new figure 3 it would be helpful, in the top panel, if the word 'noise exposure' did not spread over into the black area of the figure which represents 'no noise'.

Line 181. Change 'Controls fish' to 'Control fish'.

Line 196. Remove apostrophe in GLM's.

Line 199 onwards. I may have misunderstood the analysis for this GLMM, but was time/day not also included as an explanatory variable in this test? I assume it was, due to the reference to repeated measures on fish at different time points. If time/day was a variable in the model this should be added to the text. However, in addition, could the authors clarify how they accounted for the fact that the effect of day is very non-linear (as shown in Fig3).

Line 209. Correct 'explanatory variables' to 'explanatory variable'.

Line 214. Should fish ID be described as a random factor rather than a fixed factor?

Line 228. GLMM result – standardise the number of decimal places as with other results.

Line 238. GLM result – standardise the number of decimal places as with other results.

Line 243. The graph of parasite count through time should be figure 4, not figure 3 (references to this figure in the text also need changing).

Reviewer: 4

Comments to the Author(s)

The extra clarification and in particular the extra experiment to investigate the effect of noise on the parasites is a really great addition to this paper.

There are a couple of minor issues the authors may wish to change:

Fig 3. There is an extra space in the legend

Within the figure, there is some white over some of the "Day" labels so they read as "Dav".

L181. Control fish (not controls fish)

L210. needs a comma for clarity.

One extra thing that could be clarified - in the response to reviewers letter, the authors say that a combined effect of sound and infection likely affect mortality - but surely this interaction term has been tested in the models? This needs clarifying I think.

Finally, I would consider adding a sentence explaining that the infections were removed after day 17 for ethical reasons.

===PREPARING YOUR MANUSCRIPT===

- one version identifying all the changes that have been made (for instance, in coloured highlight, in bold text, or tracked changes);
- a 'clean' version of the new manuscript that incorporates the changes made, but does not highlight them. This version will be used for typesetting.

===PREPARING YOUR REVISION IN SCHOLARONE===

Author's Response to Decision Letter for (RSOS-200172.R1)

See Appendix B.

Decision letter (RSOS-200172.R2)

Dear Mr Masud,

It is a pleasure to accept your manuscript entitled "Noise pollution: acute noise exposure increases susceptibility to disease and chronic exposure reduces host survival" in its current form for publication in Royal Society Open Science.

You can expect to receive a proof of your article in the near future. Please contact the editorial office (openscience_proofs@royalsociety.org) and the production office (openscience@royalsociety.org) to let us know if you are likely to be away from e-mail contact -- if

you are going to be away, please nominate a co-author (if available) to manage the proofing process, and ensure they are copied into your email to the journal.

on behalf of Dr Peter Keller (Associate Editor) and Pete Smith (Subject Editor)
openscience@royalsociety.org

Appendix A

Numair Masud
Cardiff University
School of Biosciences
Museum Avenue, Cardiff
CF10 3AX

7th June 2020

Editor-in-Chief
Professor Jeremy Sanders CBE FRS
Journal of the Royal Society Open Science

Please find below our responses to each of the reviewers suggested amendments.

Overall, we found the reviewers comments and suggested edits very helpful and we have incorporated them into the finalised manuscript. In particular, we have now added a timeline diagram regarding duration of noise exposure in relation to experimental infections, as suggested by reviewers 2 and 3. In addition, we have added a power spectral density diagram in response to the suggestions of reviewer 2. We would also like to clarify the definition of what pollution is in response to reviewer 2's suggestion that using the term 'noise pollution' is not appropriate for our paper title. A pollutant can be defined as any substance or energy introduced into the environment that has a negative effect. As we have shown that noise exposure in this study does have a negative impact on disease resistance and mortality rates, we believe using the terms 'noise pollution' is appropriate. For all other minor amendments, please find details below, with our responses to each point **highlighted in red**.

We would like to add Dr Stephen Griggs as a co-author to our finalised manuscript as Dr Davide Crivelli has now left Cardiff University and Dr Grigg's has effectively taken over from Dr Crivelli's research contribution; specifically here noise file generation, hydrophone data acquisition and plotting of the power spectral density diagram.

Kind regards

Numair Masud (on behalf of all authors)

P.S. Please note that line numbering may vary depending on the version of Microsoft Word that is being utilised

Reviewer comments to Author:

Reviewer: 1

Comments to the Author(s)

This is a very well written paper on an important topic in fish husbandry that justifies publication. Minor comments:

Figure 1 legend - maintain a space between 1 1 as it looks like 11.

- Amended

Results text, first sentence - The first figure referred to relates to peak parasite burden which is Figure 2B, therefore some adjustment is needed in the figure and the second mention of the figure in the next sentence.

- Amended

A very interesting discussion. Are there any studies on the response of ectoparasites to sound waves in water or vibration?

- The only studies that have investigated how ectoparasites might be negatively impacted by sound waves, used ultrasonic pulses to induce cavitation as a potential treatment for marine sea lice (e.g. Skjelvareid et al., 2018). However, such studies used sound frequencies several orders of magnitude greater than ours and therefore a direct comparison is not possible. We have now added this to our discussion (lines 288-292).

Reviewer: 2

Comments to the Author(s)

This study provides a topical approach in understanding the effects of acoustic disturbance on fish health. The effects of noise on tertiary responses relating to fish health is largely understudied, and this work would contribute to improving the welfare of captive fish populations, and benefit the field as a whole. I'm generally positive about the study. However, I have a number of key reservations which should be addressed before being considered for publication.

The lack of formal sound analysis and presentation of acoustic data (treatments) in the manuscript is a concern. It's vital to provide full details of the stressor applied throughout the experiment. At the very least, I would like to see sound pressure levels characterised and presented for the different playback treatments. These sound levels should be calculated over the likely hearing range of the study species; presentation of mean sound levels over a wide bandwidth is misleading. Without such information it's unknown whether the levels presented were ecologically relevant, as well as precluding replication of the study.

White noise has a flat power spectrum, however playback in small tanks may drastically alter this through reflections and reverberations. As such, it would be best practice to show how the sound was altered. Further, I would like to see how the sound treatments compared to the background noise of aquaria and/or holding tank acoustic conditions. In addition, sound

pressure represents only one element of the sound field, and mention of particle motion should be included. Granted, the size and expense of accelerometers may prevent their use, but it's highly likely that guppies will be responding to particle motion than pressure. Reference should be made to particle motion even if it wasn't able to be measured at this time.

- We agree with the points raised by the reviewer regarding the noise spectrum. We have provided the PSD of the noise as measured with our hydrophone (Figure 2). We also appreciate that fish might respond to particle motion and have now mentioned this in lines 130-132 of the amended manuscript.

The manuscript is lacking in detail which made it difficult to follow in places. More clarity is needed on the exact noise exposure regime. Line 105, states chronic noise consists of 7 days of exposure, but L144–L147 counters this stating chronic noise lasted a total duration of 24 days. Perhaps an additional figure showing the experimental design would be beneficial, detailing when the noise exposure started for each treatment, when parasites were added, and how the sampling regime was undertaken.

- We have now added a timeline diagram in Figure 3 detailing length of noise exposure and duration of infections.

The methods need more detail describing exactly how all response measures were collected. Mortality as a metric first appears in the results, so comes out of the blue after reading the methods.

- A fair point. Mortality data acquisition has now been added as a detail in lines 184-187.

The statistical analysis section is missing detail on if and how model assumptions were checked in line with use of GLMs and GLMMs. It would be useful to present the full model outputs of each analysis in tables.

We have now made the suggested amendments. We now state in lines 197-198 how the error families for the GLM's were chosen. The explanatory variables for the GLM have also been added in lines 196-197. Similarly, for the use of a GLMM, lines 200-201 now states that because hosts were screened at multiple time points a GLMM was chosen to prevent pseudo-replication. Furthermore, we state in lines 190-191 that models with the lowest AIC value were chosen and that model refinement was also conducted for the GLMM. As we have also stated outputs within the result section that includes Z values, standard error and P values, we do not believe a table of outputs would add any value.

The use of noise pollution in the title and throughout the manuscript should be amended. It's difficult to link it to noise pollution in general as playback tracks all used white noise. As such, I think it more appropriate to frame the work and stimulus as an acoustic disturbance or, simply, noise exposure.

- While we appreciate the reviewer's feedback here, we argue that noise pollution is by definition acoustic disturbance. Therefore, in this instance we would prefer to retain the title as it is.

Additional comments:

L58: Range of tertiary responses have been shown with regards to fish behaviour. Suggest amend to reflect that very limited work on disease resistance in particular.

- Amended now in lines 59-60.

L100–101: Want to know sizes (standard length and mass) of fish in each treatment group, and tests to show all the same, with standard error presented.

- We do not collect data for fish mass (as this would have required additional handling of the fish which we specifically wanted to avoid), but we have now reported on the size range of fish within each treatment (lines 106-108). We also show in the statistical analysis section (with standard error presented) that standard length of fish is not a significant explanatory variable for rates of parasite increase or area under curve.

L118: UW-30s have a frequency response of 100–10,000Hz.

- Amended

L121–122: Clarification needed on what exactly this means. Was the speaker on, but playing nothing, or completely switched off and disconnected from the battery? This detail only appears in the figure 1 legend.

- Amended in lines 123-124

L180–181: Chi-squared test used to analyse host mortality, but then host mortality data in the statistics was analysed with a GLM (L197).

- Amended

L189–197: Appears that references to figures at the end of the within-text stats is in the wrong order. I.e. Area Under the Curve is made in reference to figure 2B, the bar plot of peak parasite count.

- Amended

Figure 2: Needs mention of control groups in figure legend.

- Amended

L211: Need for references showing acoustic disturbance can act as a stressor (primary, secondary mechanisms).

- We think the reviewer here is referring to the statement:

‘Acute stress has been linked to increased lymphocyte trafficking and expressions of protein cytokines from leukocytes, whereas chronic stress is associated with reduced leukocyte function (43,44). Chronic stress is generally accepted as detrimental to immunity and acute stress as potentially adaptive (43).’

And yet this is referenced so not quite sure what else is required – sorry!

L244–245: Number of papers have shown detrimental impact on at least mortality (e.g. Nedelec et al., 2015, Scientific Reports, 4, 5891). Suggest revision of this statement to include caveats, and directly tie in to the results found in this study.

- Amended

Reviewer: 3

Comments to the Author(s)

This is an interesting manuscript on a timely topic that is clearly presented. I think that the findings are important and worthy of publication in Royal Society Open Science. In my view, the authors need to clarify some details of the research protocols and (importantly) reconsider one area of their conclusions.

The authors have investigated the impact of underwater noise on parasitism in guppies. They report that acute exposure to noise before infection resulted in the development of significantly greater parasite burdens relative to controls. In contrast, fish in a treatment exposed to chronic noise throughout the experiment developed lower parasite burdens but died earlier. I found the first result especially interesting: it is striking to see that a transient exposure to 24hrs noise resulted in higher burdens developing around 2 weeks later.

Comment on the conclusions:

The most important concern I have about the manuscript is the interpretation of the results from the chronic sound treatment. Fish in this chronic sound treatment developed lower parasite burdens than fish in the control treatment, yet the fish died earlier. The authors put forward an interesting hypothesis that the chronic sound treatment may upregulate fish immune responses, that these immune responses help to reduce parasite burden, but that the responses themselves are costly and these immune costs lead to shortened lifespan. This hypothesis is worth reporting, yet it is not the only hypothesis consistent with the data. I think it is important for the authors to consider alternative hypotheses in the text, to acknowledge more clearly that this hypothesis is rather speculative (in the absence of any data about immune system activity etc), and to reconsider (reduce?) the prominence of this hypothesis in the abstract.

- We agree this is an alternate hypothesis. We have now added this to the discussion section (lines 285-292). However, as our *in vivo* investigations suggest that noise exposure does not impact parasite infrapopulations (see our response below), we feel we have not overstated our hypothesis in the abstract (lines 29-30) and that we are ‘..demonstrating a potential functional trade-off between improved parasite-resistance and shorter life span.’ In this instance, we would like to keep this line in our abstract as the most plausible hypothesis while acknowledging the reviewers alternative hypothesis in the discussion.

The reason for my concerns is based on my understanding of the experimental design. In this chronic noise treatment fish were exposed to noise for 6 days, then infected with 2 parasites on the 7th day (during which time noise exposure continued), then noise exposure continued for

a further 17 days. This treatment is compared to the control treatment (where fish were not exposed to noise), and the acute treatment (where fish were exposed to noise for 24 hours, the noise was stopped, and the fish were then infected with parasites).

For me, the key difference between the chronic and the acute treatments (which the authors do not mention in the manuscript) is that in the chronic treatment the parasites were exposed to the noise treatment (in addition to the fish), whereas in the acute treatment the parasites were never exposed to noise. This means that the authors cannot exclude the possibility that the low parasite burden in the chronic noise treatment might be caused by the direct impact of noise on the parasite (rather than the parasites being indirectly affected via an immune response putatively upregulated in the host in response to chronic noise – as the authors suggest).

Clearly there is no evidence that the parasites are indeed directly affected by chronic noise (and if they are, this could be a very important and interesting observation – of potential fundamental and applied significance). However, it strikes me that this direct effect of sound on the parasite is no less likely than the hypothesis the authors put forward (indeed impacts of noise on other invertebrate animals are known from other studies).

To summarise, the comparison between the acute and control treatment is relatively straightforward to interpret (as the authors have done), because only the fish experienced the noise stress. Whereas in the chronic noise treatment both the parasite and the fish experienced the noise, therefore the authors need to consider how effects of noise on both organisms could have driven the observed results.

The reviewer makes a valid point regarding the possibility that ectoparasites may be impacted by directly by sound exposure. However, the only research showing that sound exposure can impact ectoparasites utilised ultrasonic waves that are of frequencies several orders of magnitude higher than those we used (e.g. Skjelvareid et al., 2018). Moreover, these studies demonstrated that even at ultrasonic frequencies, sound only impacts ectoparasites when at close range to the emitted sound.

We did test the *in vivo* impact of acute (24h) noise exposure on *G. turnbulli*. This was specifically to determine whether there was any immediate effect of noise exposure on the parasites and their reproductive potential. Briefly here, n=10 fish (same sex and strain) were infected with 15 parasites each and exposed to 24 h of noise. Controls fish (n=10) were also infected but not exposed to noise. Over the 24 h time period, fish were removed and screened at two different time points (2h and 24h; so, at much shorter intervals than reported in the actual manuscript) to observe whether parasites were being removed and/or killed by noise exposure. A GLMM with a Poisson error family was utilised to analyse parasite count over time to prevent pseudo-replication as fish were screened at two time points. The explanatory variable for this model being ‘treatment’ (i.e. noise exposure versus no-sound) and fish ID being a fixed factor. For both noise exposure and controls, parasite numbers increased and there was no significant difference in total parasite count over the 24 h time period (GLMM: Z=-0.43, SE=0.06, P=0.66). We have now included this additional experiment into our manuscript in section 2.3. and reported on the results as well.

Other points for clarification etc:

Methods:

The methods section is perfectly well written, but I found the precise details of the timeline of the experiment slightly difficult to get straight in my head when I read the methods. Did the chronic exposure start 6 days before the acute exposure, then both groups infected on the same day? A timeline diagram in supplementary materials would help here, combined with some changes in the text.

- We have now added a timeline in Figure 3 in the main text to explain time of exposure and infections for both acute and chronic treatments.

Is there any difference in mean age of the fish at the time of infection driven by the timing of the sound treatments – if so, some comments would be useful.

- All fish (mixed ornamental strain) were purchased from a supplier, therefore ascertaining their age was not possible.

Can the authors clarify why the fish were treated to remove infection on day 17?

- This is in accordance with our Home Office Licence PPL 303424 and previous research conducted in our laboratory showing the typical timeline of *G. turnbulli* infections (e.g. van Oosterhout et al., 2003) and in line with the 3R's.

I think the fact that infection and sound treatment stopped on day 17 potentially complicates the interpretation of subsequent mortality observations.

- This has now been clarified by explicitly stating in the main document (lines 184-187) that deaths only occurred during infection trajectories and no subsequent mortalities occurred post-treatment.

Results:

The presentation of the infection data is very clear (and interesting). However, the results for differences in lifespan between treatments are covered in a single sentence with no figure. Would the authors consider plotting a survival curve for the three treatments, even if this ends up as a supplementary figure in the final paper? I would guess that all the fish died during the experiment, but this is not stated explicitly – could the authors clarify this. If all the fish did not die, survival analysis would be more appropriate than the GLM reported).

- This is correct. Death of fish only occurred during infection trajectories and none during post-treatment recoveries.

Could the authors comment on (and/or include appropriate analysis) what influence selective disappearance of individuals could potentially have on the parasite infection metrics? I note that average death date is around day 12 / day 14 which is about the same time that compelling treatment differences in infection metrics arise (figure 2). If deaths were non-random with respect to infection status then this might influence interpretation of the results.

- As deaths only occurred during infections, they were indeed non-random. We have now stated in the result section that days on which mortality occurred is significantly

associated with peak parasite counts and Area Under the Curve analysis (lines 303-304). Nonetheless, fish from the chronic noise treatment were dying earlier than the acute exposure and control treatments which suggests a combined effect of sound and infection on mortality.

Appendix B

Numair Masud
Cardiff University
School of Biosciences
Museum Avenue, Cardiff
CF10 3AX

20th August 2020

Editor- in- Chief
Professor Jeremy Sanders CBE FRS
Journal of the Royal Society Open Science

Overall, we have found the suggested amendments very helpful and this has improved the manuscript. In particular, we agree with Reviewer 3's suggestion of removing the suggested explanation regarding a trade-off between disease resistance and mortality. We have also added a GLMM analysis to further elucidate the effect mortality had on infection trajectories as suggested by Reviewer 3.

For all amendments please find our responses in red below. Please note that line numbering may vary based on the version of Microsoft Word being utilised. Do not hesitate to contact me should you have any further queries.

Kind regards,

Numair Masud (on behalf of all authors)

Reviewer: 2
Comments to the Author(s)

The authors have done a great job in responding to the comments raised. This has resulted in a much improved manuscript that will be a valuable addition to the field.

Thank you.

There is, however, one additional point which should be addressed before publication. The authors have now clarified that the speakers in the control trials were turned off. This creates a potential confound, whereby a magnetic field is present in the treatment trials but not the control trials. I suggest a single sentence is added in the discussion or methods to highlight this.

Amended. We have now added this possible confounding effect to our methodology (lines 123-126) where we state *"We note, however, the possibility of a confound in relation to the control fish not being exposed to magnetic fields. This is due to electrical currents creating a variable magnetic field to which the voice coil in the underwater speaker responds to generate sound"*.

Reviewer: 3

Comments to the Author(s)

I am pleased to see the changes that the authors have made since the previous round of review – I think this has resulted in an improved manuscript, which in my opinion is now close to being ready for acceptance. Almost all my comments in this review concern the difference that the authors describe between the control treatment and the chronic sound treatment. My comments outline that there are probably plausible alternative hypotheses to the hypothesis the authors favour in their discussion and abstract.

I must admit to being concerned that the authors have striven to retain statements in the paper that unambiguously claim to demonstrate that chronic sound stimulates immune responses and that investment in these immune responses shortens fish lifespan. I don't think there is any strong evidence in the paper that clearly demonstrates that this is true. To address this point the authors need to do little more than change one line in the abstract and one line in the discussion. With or without this claim, the authors still have a very interesting paper, and the fact that the other sound treatment (acute noise before parasite infection) has such an impressive effect on infection load makes the paper exciting even without the claim that the authors have retained in the manuscript regarding immune upregulation and life history costs.

We agree with the reviewer that removing the suggested explanation about the functional trade-off between disease resistance and mortality in our paper does not take away from the overall story. We have now removed this entirely.

To address this point following the previous review, the authors have included new experimental data and made alterations to the discussion.

The new data are from an experiment delivering an acute 24 hour exposure of sound to parasites on hosts; the data verify that parasites do not drop off or die during this 24 hour period. However, the new data do not provide a comparator treatment for the chronic sound treatment in the main paper where fish sites were exposed to sound for seven days. I find the argument that this new data set demonstrates that there is "no direct impact of sound on the parasites" (line 231) unconvincing because in this new experiment the parasite count was only carried out during the 24 hour time period when the fish were receiving the sound treatment (with no monitoring thereafter). This new experiment provides interesting data for the authors to include and the extra information adds to the paper. However, the result cannot be used to argue that (compared to controls) the difference in parasite numbers in the chronic exposure treatment (which lasted 14 days) can only be due to physiological changes in the fish (and not potential direct effects on the parasites).

Agreed. We have now slightly modified our results to state (lines 232-234) "*When parasites were directly exposed in vivo to noise, the number of worms alive did not significantly change over the 24 h time period (GLMM: $Z=-0.43$, $SE=0.06$, $P>0.05$) suggesting no immediate impact*

of sound exposure on the parasites". This now acknowledges that we can only conclude impacts on parasite numbers over the acute 24 h period and not comparable to chronic noise exposure. Furthermore, we have also added in our discussion that "We cannot exclude the fact that direct exposure to sound might also disrupt parasite development or cause them to actively move off the host. However, our in vivo investigations suggest that, at least over 24 h, noise exposure has no immediate impact on parasite infrapopulations." (lines 291-294).

I am content that the authors have added some well explained text to the discussion to mention the alternative hypothesis that there could be direct effects of chronic sound on the parasites. The authors now outline the possibilities that chronic sound directly impacts parasites to cause lower burdens vs the possibility that chronic sound activates the fish immune system to cause chronic burdens. However, these text changes then lead the authors to write a new concluding sentence in this section of the discussion (lines 295-298) that "While exposure to noise has been shown to impact fish mortality this is the first study to demonstrate how increased disease resistance linked to chronic noise exposure reduces survival". This is simply not true – the authors have not demonstrated that disease resistance increases in fish that receive chronic noise exposure (although lower parasite burdens might be consistent with this), furthermore they have certainly not demonstrated that this putative change in resistance is the factor that reduces fish survival (other direct effects of sound on the fish are possible). I also think it is disappointing that there has been no change in wording to the abstract in the new manuscript in relation to this issue. The current wording says that differences between the chronic sound and control treatments "demonstrating a potential functional trade-off between improved parasite resistance and shorter lifespan" doesn't really acknowledge the uncertainty or speculative nature of this claim.

Agreed. We acknowledge the speculative nature of the trade-off explanation and we have entirely removed this from our discussion.

The authors have made a brief response to my previous review comment about whether non-random mortality might cause selective disappearance of fish from the data set that might influence the results. The key issue I was raising was that if those fish that have the highest parasite burdens are most likely to die in the chronic infection treatment, then the fact that mean parasite burden is lower in this treatment could possibly be driven by the fact that the highly infected fish have died, thereby causing a low mean parasite burden at later timepoints. This issue is especially important because the chronic treatment has the highest/earliest mortality rate, thus the potential effect of selective disappearance would be greatest in this treatment. The authors include new analysis which makes me more concerned about this issue, not less concerned. The new analysis that the authors present indicates that mortality probably is being driven by infection - it appears that those fish that die earliest are those that have highest parasite burdens. This reinforces an additional potential alternative explanation for the low parasite burdens in the chronic treatment - now there are 3: (i) chronic sound upregulates an immune response that protects the fish from infection (ii) chronic sound directly negatively affects the parasite and (iii) chronic sound lowers fish tolerance of infection pathology resulting in fish with high burdens dying early.

I would encourage the authors to address this issue briefly in the results section, with appropriate modifications to the discussion section where necessary. The authors could do some of the following things (or there may be other approaches):

1. Perhaps the best way to address selective disappearance in data sets like this is to include an additional variable for 'age at death' in the statistical model, which then leaves other variables in the model to assess effects on parasite metrics once selective disappearance has been accounted for.

Agreed. Using time of death as an additional independent variable for our GLM was already included in the original amended manuscript and we showed that time of death did significantly impact overall infection trajectories (lines 235-238).

2. The authors might also present the data for the days on which the fish died, rather than just the mean death days (I suggested they add a survival curve to the manuscript in my previous review). This might allow them to argue that (perhaps) substantial differences in mean parasite characteristics emerge in (old)fig3 before any mortality has occurred.

Please see response immediately below Point 3.

3. The authors could reanalyse data including only those fish that survive all the way through to (eg) day 15, if the difference between the chronic and control treatments still remains then then authors could conclude that selective disappearance isn't important/the-only-factor driving the difference seen.

Point 3 (above) is a better way of analysing the impact of selective disappearance than point 2 above – thank you for the suggestions. We have now conducted a further analysis on only those fish surviving until day 17. This clearly shows that the significance is no longer seen between chronic noise and control fish parasite counts ($Z=-1.4$, $SE=0.88$, $P>0.05$). We have now added this analysis in our methodology (lines 217-219) and results (lines 238-241). We have also adjusted our discussion accordingly to state that *"In our study, chronic noise exposure was linked to significantly earlier host mortality and reduced pathogen burdens compared to fish from acute noise and control treatments, which strongly indicates that chronic noise reduces pathogen tolerance"* (lines 288-291).

It may be that these analyses mean that the authors can conclude that selection disappearance is not important, which would simplify any changes required.

Other comments:

Lines 68-70 – introduction. In reference to Wysocki et al. it seems odd to phrase this sentence that this is the only study to 'demonstrate an unambiguous effect', when the sentence goes on to say that fish in this study were 'unaffected by chronic 8-month noise exposure'. It is not clear to me what the "unambiguous effect" is.

A fair point. We have now slightly rephrased this to state “*only Wysocki et al. (20) demonstrated that rainbow trout (*Oncorhynchus mykiss*) appeared unaffected by chronic 8-month noise exposure and subsequent *Yersinia ruckeri* inoculation*”. We explicitly avoided further comment on reference 19 in the introduction because having revisited their methods (specifically Chi-squared tests), it was apparent that there was an error in their analysis, but we felt our paper was not the place to dissect this.

On the new figure 3 it would be helpful, in the top panel, if the word ‘noise exposure’ did not spread over into the black area of the figure which represents ‘no noise’.

Amended.

Line 181. Change ‘Controls fish’ to ‘Control fish’.

Amended.

Line 196. Remove apostrophe in GLM’s.

Amended.

Line 199 onwards. I may have misunderstood the analysis for this GLMM, but was time/day not also included as an explanatory variable in this test? I assume it was, due to the reference to repeated measures on fish at different time points. If time/day was a variable in the model this should be added to the text. However, in addition, could the authors clarify how they accounted for the fact that the effect of day is very non-linear (as shown in Fig3).

Yes - there does seem to be a misunderstanding here regarding our GLMM analysis. We did not include day in the analysis as for this host-parasite system (guppy- *Gyrodactylus turnbulli*), day will always be a significant variable statistically as reproduction of this pathogen is exponential. Therefore, adding day into our model would not add any biological significance to our results other than stating that day does influence parasite number. Furthermore, the robustness of our model would be compromised as the variable ‘day’ would mask the effects of other independent variables. This may also relate to a misunderstanding that the reviewer brought up below where they suggest we should treat fish ID as a fixed factor rather than a random factor. To clarify, a fixed factor is one where we want to know what effect it is having on the dependant variable (i.e. parasite count), whereas a random variable is one that is being introduced to prevent pseudo-replication (see response below).

The non-linearity of infection trajectories (Figure 4 in manuscript) is largely explained by host mortality being a significant explanatory variable for parasite numbers. Selective disappearance of hosts would influence the linearity of the trajectory. Indeed, the trajectories for all three treatments begin to level-off after day 12 for chronic exposure and day 14 for acute exposure and controls which corresponds to average death days for all treatments.

Line 209. Correct ‘explanatory variables’ to ‘explanatory variable’.

Amended.

Line 214. Should fish ID be described as a random factor rather than a fixed factor?

Yes, all our model specifications are correct. As fish were screened at multiple time-points, if we do not specify in our model that fish ID is a random variable, R would analyse each data point (i.e. day on which screening occurred) as a separate fish and this would introduce pseudoreplication. Therefore, we must treat fish ID as a random variable.

Line 228. GLMM result – standardise the number of decimal places as with other results.

Amended.

Line 238. GLM result – standardise the number of decimal places as with other results.

Amended.

Line 243. The graph of parasite count through time should be figure 4, not figure 3 (references to this figure in the text also need changing).

Amended.

Reviewer: 4

Comments to the Author(s)

The extra clarification and in particular the extra experiment to investigate the effect of noise on the parasites is a really great addition to this paper.

There are a couple of minor issues the authors may wish to change:

Fig 3. There is an extra space in the legend

Within the figure, there is some white over some of the "Day" labels so they read as "Dav".

Amended.

L181. Control fish (not controls fish)

Amended .

L210. needs a comma for clarity.

Amended.

One extra thing that could be clarified - in the response to reviewers letter, the authors say that a combined effect of sound and infection likely affect mortality - but surely this interaction term has been tested in the models? This needs clarifying I think.

Perhaps the reviewer is still thinking of the original manuscript here. This is a point we clarified in the first amendment of the manuscript, where in lines 188-189 we altered the text to read “*Fish mortality only occurred during infections.*” No mortality occurred in sound only treatments that were not infected. In addition, fish exposed to chronic noise died significantly earlier than acute and no noise treatments when under infections, which suggests that infection on its own clearly has an impact on mortality, and when combined with chronic noise, mortality occurs earlier.

Finally, I would consider adding a sentence explaining that the infections were removed after day 17 for ethical reasons.

Day 17 was chosen based on our previous *G. turnbulli* infections and also in accordance with the animal welfare principles of the 3R's and reviewed by Cardiff Universities Animal Ethics Committee. We had already stated in our previous manuscript (lines 313-314) that all animal welfare practice is in accordance with Home Office licencing and the Animal Ethics Committee. We have now added a new line (line 164) in our methodology stating, “...*a timeline determined from our previous G. turnbulli infections (e.g. van Oosterhout et al., 2003)*”.